# Graph Coverings for Investigating Non Local Structures in Proteins, Music and Poems

**Michel Planat** [1,*,†] , **Raymond Aschheim** [2,†] , **Marcelo M. Amaral** [2,†] , **Fang Fang** [2,†] and **Klee Irwin** [2,†]

1   Institut FEMTO-ST CNRS UMR 6174, Université de Bourgogne/Franche-Comté, 15B Avenue des Montboucons, F-25044 Besançon, France
2   Quantum Gravity Research, Los Angeles, CA 90290, USA; Raymond@QuantumGravityResearch.org (R.A.); Marcelo@quantumgravityresearch.org (M.M.A.); Fang@QuantumGravityResearch.org (F.F.); Klee@quantumgravityresearch.org (K.I.)
*   Correspondence: michel.planat@femto-st.fr
†   These authors contributed equally to this work.

**Abstract:** We explore the structural similarities in three different languages, first in the protein language whose primary letters are the amino acids, second in the musical language whose primary letters are the notes, and third in the poetry language whose primary letters are the alphabet. For proteins, the non local (secondary) letters are the types of foldings in space ($\alpha$-helices, $\beta$-sheets, etc); for music, one is dealing with clear-cut repetition units called musical forms and for poems the structure consists of grammatical forms (names, verbs, etc). We show in this paper that the mathematics of such secondary structures relies on finitely presented groups $f_p$ on $r$ letters, where $r$ counts the number of types of such secondary non local segments. The number of conjugacy classes of a given index (also the number of graph coverings over a base graph) of a group $f_p$ is found to be close to the number of conjugacy classes of the same index in the free group $F_{r-1}$ on $r-1$ generators. In a concrete way, we explore the group structure of a variant of the SARS-Cov-2 spike protein and the group structure of apolipoprotein-H, passing from the primary code with amino acids to the secondary structure organizing the foldings. Then, we look at the musical forms employed in the classical and contemporary periods. Finally, we investigate in much detail the group structure of a small poem in prose by Charles Baudelaire and that of the *Bateau Ivre* by Arthur Rimbaud.

**Keywords:** protein structure; musical forms; poetry; graph coverings; finitely generated groups; SARS-Cov-2; Arthur Rimbaud





## 1. Introduction

In this paper, we point out for the first time a remarkable analogy between the pattern structure of bonds between amino acids in a protein (the protein secondary structure [1]) and the non local structures observed in tonal music and in poems. We explain the origin of these analogies with finitely generated groups and graph covering theory.

A protein is a long polymeric linear chain encoded with 20 letters (the 20 amino acids). The surjective mapping of the $4^3 = 64$ codons to the 20 amino acids is the DNA genetic code. It can be given a mathematical theory with appropriate finite groups [2,3]. In addition, a protein folds in the three dimensional space with structural elements such as coils, $\alpha$-helices and $\beta$-sheets, or other arrangements that determine its biological function. The number of proteins encoded in genomes depends on the biological organism (typically from 1 to $10^2$ proteins in viruses, from $10^2$ to $10^3$ proteins in bacteria and from $10^3$ to $10^4$ proteins in eukaryotes). The protein database (or PDB) contains about $1.8 \times 10^5$ entries [4]. Proteins ensure the language of life, amino acids are the alphabet, proteins are the words and the set of proteins in an organism are the phrases.

Analogously in music, a note is a letter encoding a musical sound. In the 12-tone chromatic scale [5], each of the 12 notes (or letters) has the frequency of the previous note

multiplied by $2^{1/12} \approx 1.0595$. The form refers to the secondary structure of a musical composition in terms of clear-cut units of equal length, for example, A-B-A in the sonata form or A-B-C-B-A in an arch form [6].

Now, we come to human language and the Latin alphabet. There are 26 letters organized into words of various types such as names, adjectives, verbs, and so on. In the following, we will show that a verse in a poem or a phrase in prose have distinctive features, the former being closer to our theory.

Our mathematical theory of the secondary structures in proteins, music and poems relies on the concept of a finitely generated group and the corresponding graph coverings, as explained in Section 2.

We will investigate three applications of the graph covering approach. In Section 3, we look at the secondary structures of two proteins. We take as examples the spike protein of the SARS-Cov-2 virus and a glycoprotein playing a role in the immune system (see [3] for our earlier work). In Section 4, the secondary structures are the musical forms of western music in the classical age and twentieth century music. Then, in Section 5, the secondary structures in the verses of selected poems are obtained from an encoding of the types of words (names, verbs, prepositions, etc).

*A Brief Review of the Literature*

After we received an invitation to contribute to the present special issue of Sci "Mathematics and poetry, with a view towards machine learning" we thought that our current group theoretical approach of protein language [3] could be converted into an understanding of the poetic language, as well as an understanding of some musical structures.

Our goal in this subsection is to point out earlier work in the same direction as ours. There are many papers attempting to relate group theory to the genetic code, as reviewed in [2] but we found none of them featuring the secondary structure of proteins along the chain of amino acids, as we did in [3] and as we do below with the graph coverings.

Poetry inspired mathematics has been the common thread of most papers exploring the connection between poems and maths [7–10]. However, it is more challenging to explain what type of structure and beauty occurs in a poem in the language of mathematics [11]. Perhaps mathematical linguistics is the proper frame for making progress [12] and artificial intelligence (AI) may help in the classification of languages [13].

Although both subjects have been connected for centuries, comparing musical structures to mathematics is a fairly new research domain [14]. For a different perspective, the readers may consult Reference [15].

## 2. Graph Coverings and Conjugacy Classes of a Finitely Generated Group

Let $\mathrm{rel}(x_1, x_2, \ldots, x_r)$ be the relation defining the finitely presented group $fp = \langle x_1, x_2, \ldots, x_r | \mathrm{rel}(x_1, x_2, \ldots, x_r) \rangle$ on $r$ letters (or generators). We are interested in the conjugacy classes (cc) of subgroups of $fp$ with respect to the nature of the relation rel. In a nutshell, one observes that the cardinality structure $\eta_d(fp)$ of conjugacy classes of subgroups of index $d$ of $fp$ is all the closer to that of the free group $F_{r-1}$ on $r-1$ generators as the choice of rel contains more non local structure. To arrive at this statement, we experiment on protein foldings, musical forms and poems. The former case was first explored in [3].

Let $X$ and $\tilde{X}$ be two graphs. A graph epimorphism (an onto or surjective homomorphism) $\pi : X \to \tilde{X}$ is called a covering projection if, for every vertex $\tilde{v}$ of $\tilde{X}$, $\pi$ maps the neighborhood of $\tilde{v}$ bijectively onto the neighborhood of $\pi\tilde{v}$. The graph $X$ is referred to as a base graph (or a quotient graph) and $\tilde{X}$ is called the covering graph. The conjugacy classes of subgroups of index $d$ in the fundamental group of a base graph $X$ are in one-to-one correspondence with the connected $d$-fold coverings of $X$, as it has been known for some time [16,17].

Graph coverings and group actions are closely related.

Let us start from an enumeration of integer partitions of $d$ that satisfy:

$$l_1 + 2l_2 + \ldots + dl_d = d,$$

a famous problem in analytic number theory [18,19]. The number of such partitions is $p(d) = [1, 2, 3, 5, 7, 11, 15, 22 \cdots]$ when $d = [1, 2, 3, 4, 5, 6, 7, 8 \cdots]$.

The number of $d$-fold coverings of a graph $X$ of the first Betti number $r$ is ([17], p. 41),

$$\text{Iso}(X; d) = \sum_{l_1 + 2l_2 + \ldots + dl_d = d} (l_1! 2^{l_2} l_2! \ldots + d^{l_d} l_d!)^{r-1}.$$

Another interpretation of $\text{Iso}(X; d)$ is found in ([20], Euqation (12)). Taking a set of mixed quantum states comprising $r + 1$ subsystems, $\text{Iso}(X; d)$ corresponds to the stable dimension of degree $d$ local unitary invariants. For two subsystems, $r = 1$ and such a stable dimension is $\text{Iso}(X; d) = p(d)$. A table for $\text{Iso}(X, d)$ with small $d$'s is in ([17], Table 3.1, p. 82) or ([20], Table 1).

Then, one needs a theorem derived by Hall in 1949 [21] about the number $N_{d,r}$ of subgroups of index $d$ in $F_r$

$$N_{d,r} = d(d!)^{r-1} - \sum_{i=1}^{d-1} [(d-i)!]^{r-1} N_{i,r}$$

to establish that the number $\text{Isoc}(X; d)$ of connected $d$-fold coverings of a graph $X$ (alias the number of conjugacy classes of subgroups in the fundamental group of $X$) is as follows ([17], Theorem 3.2, p. 84):

$$\text{Isoc}(X; d) = \frac{1}{d} \sum_{m|d} N_{m,r} \sum_{l | \frac{d}{m}} \mu\left(\frac{d}{ml}\right) l^{(r-1)m+1},$$

where $\mu$ denotes the number-theoretic Möbius function.

Table 1 provides the values of $\text{Isoc}(X; d)$ for small values of $r$ and $d$ ([17], Table 3.2).

**Table 1.** The number $\text{Isoc}(X; d)$ for small values of first Betti number $r$ (alias the number of generators of the free group $F_r$) and index $d$. Thus, the columns correspond to the number of conjugacy classes of subgroups of index $d$ in the free group of rank $r$.

| r | d = 1 | d = 2 | d = 3 | d = 4 | d = 5 | d = 6 | d = 7 |
|---|-------|-------|-------|-------|-------|-------|-------|
| 1 | 1 | 1 | 1 | 1 | 1 | 1 | 1 |
| 2 | 1 | 3 | 7 | 26 | 97 | 624 | 4163 |
| 3 | 1 | 7 | 41 | 604 | 13,753 | 504,243 | 24,824,785 |
| 4 | 1 | 15 | 235 | 14,120 | 1,712,845 | 371,515,454 | 127,635,996,839 |
| 5 | 1 | 31 | 1361 | 334,576 | 207,009,649 | 268,530,771,271 | 644,969,015,852,641 |

The finitely presented groups $G = f_p$ may be characterized in terms of a first Betti number $r$. For a group G, $r$ is the rank (the number of generators) of the abelian quotient $G/[G, G]$. To some extent, a group $f_p$ whose first Betti number is $r$ may be said to be close to the free group $F_r$ since both of them have the same minimum number of generators.

## 3. Graph Coverings for Proteins

As a follow up of our previous paper [3] we first apply the above theory to two proteins of current interest, the spike protein in a variant of SARS-Cov-2 and a protein that plays an important role in the immune system.

### 3.1. The D614G Variant (Minus RBD) of the SARS-CoV-2 Spike Protein

As a first example of the application of our approach, let us consider the D614G variant (minus RBD: the receptor binding domain) of the SARS-CoV-2 spike protein. In the Protein Data Bank in Europe, the name of the sequence is 6XS6 [22]. D614G is a missense mutation

(a nonsynonymous substitution where a single nucleotide results in a codon that codes for a different amino acid). The mutation occurs at position 614 where glycine has replaced aspartic acid worldwide. Glycine increases the transmission rate and correlates with the prevalence of loss of smell as a symptom of COVID-19, possibly related to a higher binding of the RBD to the ACE2 receptor: an enzyme attached to the membrane of heart cells. A picture of the secondary structures can be found in Figure 1.

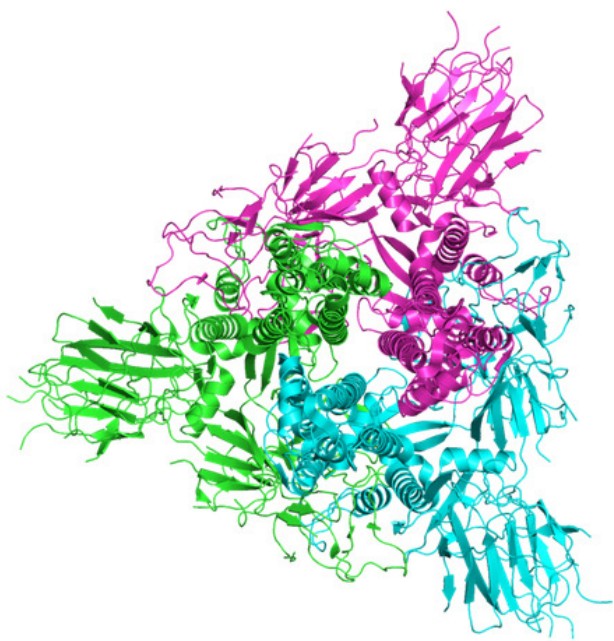

**Figure 1.** A picture of the secondary structure of D614G variant (minus RBD) of the SARS-CoV-2 spike protein found in the protein data bank in Europe [22].

The D614G variant (minus RBD) of the SARS-CoV-2 spike protein contains 786 amino acids (aa) forming a (long) word as follows:

AYTNSFTRGVYYPDKVFRSSVLHSTQDLFLPFFSNVTWFHAIHDNPVLPF…
AYRFNGIGVTQNVLYENQKLIANQFNSAIGKIQDSLSSTASALGKLQDVV.
NTQEVFAQVKQIYKTPPIKDFGGFNFSQILPDPSKPSKRSFIEDLLFNKV…
FVTQRNFYEPQIITTDNTFVSGNCDVVIGIVNNTV

Such a protein sequence, comprising 20 amino acids as letters of the primary code, can be encoded in terms of secondary structures. Most of the time, for proteins, one makes use of three types of encoding that are segments of $\alpha$ helices (encoded with the symbol *H*), segments of $\beta$ pleated sheets (encoded by the symbol *E*) and the segment of random coils (encoded by the segment *C*) [1,3,23].

A finer structure may be obtained by using methods such as the SST Bayesian method. A summary of the approach can be found in Reference [23].

We used a software prepared in [24] to obtain the following secondary structure

rel(H,E,C,G,I,T,4) =

CCCCCCCEEEEEECCCCCCCEEEEECCCCCCCCCCCCEEEEEECCCCCCCCC…
HHHHHHHHCC444444CHHHHHHHHHHHHHHHHHHHHHHHCCCGGGGGHHHHH
HHIIIIICCCCCCCCCCCCCCCCCCCCCCTTTTTCCCCCCCCCHHHHHHHHHHH…
CCCTTTTTCCCCCTTTTTCCCC44444EEEEEECC,

where *G* means a $3_{10}$ helix, 4 means $\alpha$-like turns, *I* means a right-handed $\pi$ helix and *T* corresponds to unspecified turns.

For the group analysis, we slightly simplify the problem by taking $4 = H$ just one form of $\alpha$ turn so that the sequence is encoded with 6 letters only. Then, we further simplify by taking $T = C$ to obtain a 5-letter encoding. We further simplify by taking $I = H$, then by taking $G = H$ to get 4-letter and 3-letter encodings, respectively. The results are in Table 2.

**Table 2.** Group analysis of the D614G variant (minus RBD) of the SARS-CoV-2 spike protein. The bold numbers mean that the cardinality structure of cc of subgroups of $G$ fits that of the free group $F_{r-1}$ when the encoding makes use of $r$ letters. In the last column, $r$ is the first Betti number of the generating group $f_p$.

| PDB 6XS6: AYTNSFTRGVYYPDKVFRSSVLHSTQDL . . . | Cardinality Structure of cc of Subgroups | r |
|---|---|---|
| 6 letters H, E, C, G, I, T | **[1,31,1361,334576]** | 5 |
| 5 letters H, E, C, G, I | **[1,15,235,14120]** | 4 |
| 4 letters H, E, C, G | **[1,7,41 604,**14720] | 3 |
| 3 letters H, E, C | **[1,3,7,**30,127,926] | 2 |

We observe that the cardinality structure of the cc of subgroups of the finitely presented groups $fp = \langle H, E, C, G, I, T | rel \rangle, \ldots, fp = \langle H, E, C | rel \rangle$ fits the free group $F_{r-1}$ when the encoding makes use of $r = 6, 5, 4, 3$ letters. This is in line with our results found in [3] on several kinds of proteins.

*3.2. The $\beta$-2-Glycoprotein 1 or Apolipoprotein-H*

Our second example deals with a protein playing an important role in the immune system [25]. In the Protein Data Bank, the name of the sequence is 6V06 [26] and it contains 326 aa. All models predict secondary structures mainly comprising $\beta$-pleated sheets and random coils and sometimes short segments of $\alpha$-helices.

We observe in Table 3 that the cardinality structure of the cc of subgroups of the finitely presented groups $fp = \langle H, E, C | rel \rangle$ approximately fits the free group $F_2$ on two letters for the first three models but not for the RAPTORX model. In one case (with the PORTER model [27]), all first six digits fit those of $F_2$ and higher order digits could not be reached. The reader may refer to our paper [3] where such a good fit could be obtained for the sequences in the arms of the protein complex Hfq (with 74 aa). This complex with the 6-fold symmetry is known to play a role in DNA replication.

A picture of the secondary structure of the apolipoprotein-H obtained with the software of Ref. [24] is displayed in Figure 2.

**Table 3.** Group analysis of apolipoprotein-H (PDB 6V06). The bold numbers means that the cardinality structure of cc of subgroups of $f_p$ fits that of the free group $F_3$ when the encoding makes use of 2 letters. The first model is the one used in the previous Section [24] where we took $4 = H$ and $T = C$. The other models of secondary structures with segments E, H and C are from softwares PORTER, PHYRE2 and RAPTORX. The references to these softwares may be found in our recent paper [3]. The notation $r$ in column 3 means the first Betti number of $f_p$.

| PDB 6V06: GRTCPKPDDLPFSTVVPLKTFYEPG... | Cardinality Structure of cc of Subgroups | r |
|---|---|---|
| Konagurthu | **[1,3,7,26,**218,2241] | 2 |
| PORTER | **[1,3,7,26,97,624]** | . |
| PHYRE2 | **[1,3,7,26,**157,1046] | . |
| RAPTORX | **[1,**7,17,134,923,13317] | 3 |

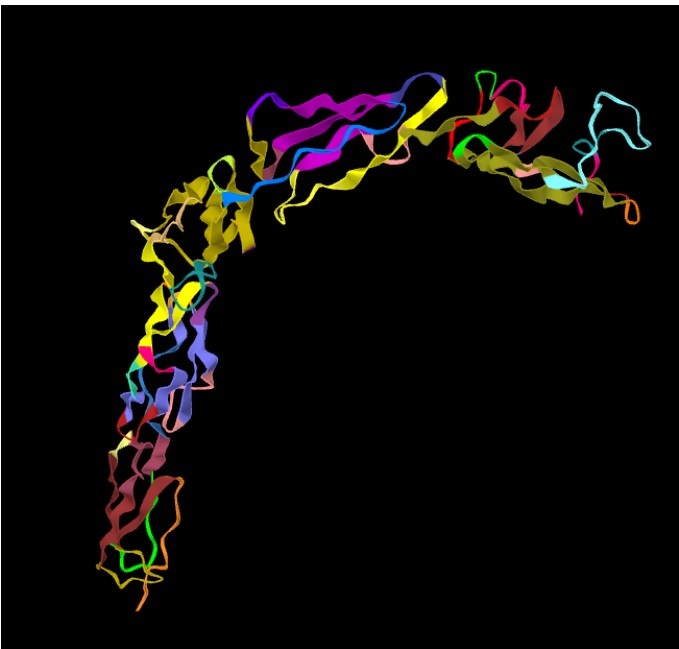

**Figure 2.** A picture of the secondary structure of the apolipoprotein-H obtained with the software [24].

## 4. Graph Coverings for Musical Forms

We accept that this structure determines the beauty in art. We provide two examples of this relationship, first by studying musical forms, then by looking at the structure of verses in poems. Our approach encompasses the orthodox view of periodicity or quasi-periodicity inherent to such structures. Instead of that and the non local character of the structure is investigated thanks to a group with generators given by the allowed generators $x_1, x_2, \cdots, x_r$ and a relation rel, determining the position of such successive generators, as we did for the secondary structures of proteins.

### 4.1. The Sequence Isoc(X; 1), the Golden Ratio and More

4.1.1. The Fibonacci Sequence

As shown in Table 1, the sequence $\text{Isoc}(X; 1)$ only contains 1 in its entries and it is tempting to associate this sequence to the most irrational number, the Golden ratio $\phi = (\sqrt{5} - 1)/2$ through the continued fraction expansion $\phi = 1/(1 + 1/(1 + 1/(1 + 1/(1 + \cdots)))) = [0; 1, 1, 1, 1, \cdots)$.

Let us now take a two-letter alphabet (with letters $L$ and $S$) and the Fibonacci words $w_n$ defined as $w_1 = S$, $w_2 = L$, $w_n = w_{n-1} w_{n-2}$. The sequence of Fibonacci words $w_n$ is as follows

$$S, L, LS, LSL, LSLLS, LSLLSLSL, LSLLSLSLLSLLS, LSLLSLSLLSLLSLSLLSLSL, \cdots$$

and its length corresponds to the Fibonacci numbers $1, 1, 2, 3, 5, 8, 13, 21, \cdots$.

Then, one can check that the finitely-presented group $f_p(n) = \langle S, L | w_n \rangle$ whose relation is a Fibonacci word $w_n$ possesses a cardinality sequence of subgroups $[1, 1, 1, 1, 1, 1, 1, 1 \cdots)$ equal to $\text{Isoc}(X; 1)$, up to all computable orders, despite the fact that the groups $f_p(n)$ are not the same. It is straightforward to check that the first Betti number $r$ of $f_p(n)$ is 1, as expected.

4.1.2. The Period Doubling Cascade

Other rules lead to a Betti number $r = 1$ and the corresponding sequence Isoc(X;1). Let us consider the period-doubling cascade in the logistic map $x_{l+1} = 1 - \lambda x_l^2$. Period doubling can be generated by repeated use of the substitutions $R \to RL$ and $L \to RR$., so that the sequence of period doubling is [28]

$$R, L, RL, RLR^2, RLR^3LRL, RLR^3LRLRLR^3LR^3, RLR^3LRLRLR^3LR^3LR^3LRLRLR^3LRLRL, \cdots$$

and the corresponding finitely presented groups also have first Betti numbers equal to 1.

### 4.1.3. Musical Forms of the Classical Age

Going into musical forms, the ternary structure L-S-L (most commonly denoted $A - B - A$) corresponding to the Fibonacci word $w_4$ is a Western instrumental genre notably used in sonatas, symphonies and string quartets. The basic elements of sonata forms are the exposition $A$, the development $B$ and recapitulation $A$. While the musical form $A - B - A$ is symmetric, the Fibonacci word $A - B - A - A - B$ corresponding to $w_5$ is asymmetric and used in some songs or ballads from the Renaissance.

In a closely related direction, it was shown that the lengths $a$ and $b$ of sections $A$ and $B$ in all Mozart's sonata movements are such that the ratio $b/(a+b) \approx \phi$ [29].

### 4.2. The Sequence Isoc$(X;2)$ in Twentieth Century Music and Jazz

In the 20th century, musical forms escaped the classical channels that were created. With the Hungarian composer Béla Bartók, a musical structure known as the arch form was created. The arch form is a sectional structure for a piece of music based on repetition, in reverse order, so that the overall form is symmetric, most often around a central movement. Formally, it looks like $A - B - C - B - A$. A well known composition of Bartok with this structure is *Music for strings, percussion and celesta* [30]. In Table 4, it is shown that the cardinality sequence of cc of subgroups of the group generated with the relation rel=$ABCBA$ corresponds to Isoc$(X;2)$ up to the higher index 9 that we could check with our computer. A similar result is obtained with the symmetrical word $ABACABA$.

Our second example is a musical form known as twelve-bar blues [31], one of the most prominent chord progressions in popular music and jazz. In this context, the notation $A$ is for the tonic, $B$ is for the subdominant and $C$ is for the dominant, each letter representing one chord. In twelve-bar blues, there are twelve chords arranged as in the first column of Table 4. We observe that the standard twelve-bar blues are different in structure from the sequence of Isoc$(X;2)$. However, variations 1 and 2 have a structure close to Isoc$(X;2)$. In the former case, the first 9 orders lead to the same digit in the sequence.

Our third example is the musical form A-A-B-C-C. Notably, it is found in the *Slow movement from Haydn's 'Emperor quartet Opus 76, N°3* [32] (Figure 3), much sooner than the contemporary period. (See also Ref. [33] for the frequent occurrence of the same musical form in djanba songs at Wadeye.) As in the aforementioned examples, the cardinality sequence of the cc of subgroups of the group built with rel=AABCC corresponds to Isoc$(X;2)$ up to the highest index 9 that we could reach in our calculations.

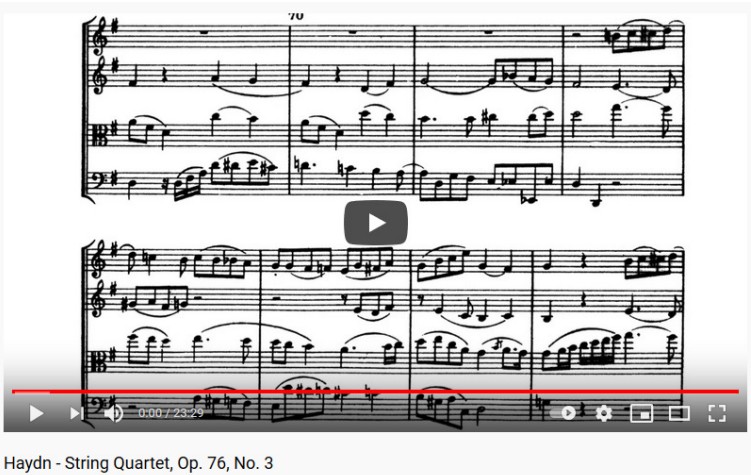

Haydn - String Quartet, Op. 76, No. 3

**Figure 3.** Slow movement from Haydn's 'Emperor' quartet Opus 76, N°3.

**Table 4.** Group analysis of a few musical forms whose structure of subgroups, apart from exceptions, is close to Isoc$(X;d)$ with $d = 2$ (at the upper part of the table) or $d = 3$ (at the lower part of the table). Of course, the forms A-B-C and A-B-C-D have the cardinality sequence of cc of subgroups exactly equal to Isoc$(X;2)$ and Isoc$(X;3)$, respectively.

| Musical Form | Ref | Card. Struct. of cc of Subgr. | r |
|---|---|---|---|
| A-B-C-B-A | arch, Belá Bartók | [**1,3,7,26,97,624,** **4163,34470,314493**] | 2 |
| . | . | . | . |
| A-B-A-C-A-B-A | . | . | . |
| A-B-A-C-A, A-B-A-C-A-B-A | rondo | . | . |
| A-B-A-C | | . | . |
| A-A-B-C-C | Haydn [32], | . | . |
| . | djanba ([33], Figure 9.8) | . | . |
| A-A-A-A-B-B-A-A-C-C-A-A | twelve-bar blues, standard | [**1**,7,14,109,396,3347, 19758,287340] | 3 |
| . | . | . | . |
| A-A-A-A-B-B-A-A-C-B-A-A | twelve-bar blues, variation 1 | [**1,3,7,26,97,624,** **4163,34470,314493**] | 2 |
| . | . | . | . |
| A-A-A-A-B-B-A-A-B-C-A-C | twelve-bar blues, variation 2 | [**1,3,7,26**,127, 799, 5168, 42879] | . |
| . | . | . | . |
| A-B-C | Isoc$(X;2)$ | [**1,3,7,26,97,624,** **4163,34470,314493**] | 2 |
| . | . | . | . |
| A-A-B-B-C-C-D-D | pot pourri | [**1,15**,82,1583,30242] | 4 |
| A-B-A-C-A-D-A | rondo | [**1,7,41,604,13753,504243**] | 3 |
| A-B-C-D | Isoc$(X;3)$ | [**1,7,41,604,13753,** **504243,24824785**] | 3 |
| . | . | . | . |

Further musical forms with 4 letters A, B, C, and D and their relationship to Isoc$(X;3)$ are provided in the lower part of Table 4.

Not surprisingly, the rank $r$ of the abelian quotient of $f_p = \langle A, B, C | \mathrm{rel}(A, B, C) \rangle$ is found to be 2 when the cardinality structure fits that Isoc$(X;2)$ in Table 4. Otherwise, the rank is 3. Similarly, the rank $r$ of the abelian quotient of $f_p = \langle A, B, C, D | \mathrm{rel}(A, B, C, D) \rangle$ is found to be 3 when the cardinality structure fits that Isoc$(X;3)$ in Table 4. Otherwise, the rank is 4.

## 5. Graph Coverings for Prose and Poems

### 5.1. Graph Coverings for Prose

Let us perform a group analysis of a long sentence in prose. We selected a text by Charles Baudelaire [34]:

> *Le gamin du céleste Empire hésita d'abord; puis, se ravisant, il répondit: "Je vais vous le dire ". Peu d'instants après, il reparut, tenant dans ses bras un fort gros chat, et le regardant, comme on dit, dans le blanc des yeux, il affirma sans hésiter: "Il n'est pas encore tout à fait midi." Ce qui était vrai.*

In Table 5, the group analysis is performed with 3, 4 or 5 letters (in the upper part) and is compared to random sequences with the same number of letters (in the lower part).

The text of the sentence is first encoded with three letters ($H$ for names and adjectives, $E$ for verbs and $C$ otherwise), we observe that the subgroup structure has cardinality close to that of a free group $F_2$ on two letters up to index 3. If one adds one letter $A$ for the prepositions in the sentence (in addition to $H$, $E$ and $C$), then the subgroup structure has cardinality close to that of a free group $F_3$ on three letters. If adverbs $B$ are also selected, then the subgroup structure is close to that of the free group $F_4$. In all three cases, the similarity holds up to index 3 and that the cc of subgroups are the same as in the corresponding free groups. The first Betti numbers of the generating groups are 2, 3 and 4 as expected.

In Table 5, we also computed the cardinality structure of the cc of subgroups of small indexes obtained from a random sequence of 250 letters (like the number of letters in the previously studied sentence of the small poem in prose). One took 10 runs with

random sequences having 3, 4 or 5 letters. We see that the cardinality structure of the cc of subgroups for the cases with 4 or 5 letters tends to align to that of the free group $F_{r-2}$ (not $F_{r-1}$). The 3-letter case is the most random one and does not correspond to $F_1$ (or $F_2$), in most runs.

Our conclusion is that the considered prose sequence contains a structure close to that of $F_r$ when we select $r + 1$ letters for the encoding of the sentence, a result that is similar to that which we found in the group analysis of proteins in Section 3 and musical forms in Section 4.

**Table 5.** Group analysis of an excerpt of a small poem in prose *Le vieux saltimbanque* by Charles Baudelaire. The text is split into segments encoded by the symbol *H* (for names and adjectives), *E* (for verbs), *A* for prepositions, *B* for adverbs, or *C* (for the other types: conjunctions, punctuation marks and so on). The cardinality structure of the cc of subgroups of a small index is compared to the one obtained with 10 runs of a sequence of words of a similar length (i.e., the length 250) with the corresponding number of letters.

| Le Gamin du Céleste Empire ... Ce Qui était Vrai. | Card. Seq. of cc of Subgroups | r |
|---|---|---|
| 3 letters: rel=$C^2H^5C^2H^7H^6E^6C^7CC^4CC^2E^8C\cdots$ | **[1,3,7,34**,131] | 2 |
| 4 letters: rel=$C^2H^5A^2H^7H^6E^6C^7CC^4CC^2E^8C\cdots$ | **[1,7,41**,636,14364] | 3 |
| 5 letters: rel=$C^2H^5A^2H^7H^6E^6B^7CB^4CC^2E^8C\cdots$ | **[1,15,235**,14376,.] | 4 |
| [Random[1,3]: i in [1..250]] | **[1**,1,1,2,4,4] | 1 |
| (10 runs) | **[1,3**,2,9,5,20] | 2 |
|  | **[1,3**,1,6,6,15] | . |
|  | **[1,3,7**,30,124,987] | . |
|  | **[1**,7,17,126,323,2445] | 3 |
|  | etc |  |
| Isoc(X;2) | **[1,3,7,26,97,624]** | 2 |
| [Random[1,4]: i in [1..250]] | **[1,3,7**,30,.] (×3) | 2 |
| (10 runs) | **[1,3,10**,51,.] (×3) | . |
|  | **[1,3,7,26**,457] | . |
|  | **[1,3**,10,39,.] | . |
|  | **[1,3**,13,52,.] | . |
|  | **[1**,7,20,143,.] | 3 |
| Isoc(X;3) | **[1,7,41,604,13573]** | 3 |
| [Random[1,5]: i in [1..250]] | **[1,7,41**,620,.] (×3) | 3 |
| (10 runs) | **[1,7,41**,636,.] (×3) | . |
|  | **[1,7,41,604**,.] (×2) | . |
|  | **[1,7,41**,668,.] | . |
|  | **[1,7**,50,819,.] | . |
| Isoc(X;4) | **[1,15,235,14120,1712845]** | 4 |

### 5.2. Graph Coverings for Poems

In poems, the verses are generally of a smaller length than that for a sentence in prose. We selected the first strophe of the poem, *Le Bateau Ivre,* by Arthur Rimbaud. The poem may be found on a wall in Paris, see Figure 4. The verses in the strophe have about 35 letters. We compare the group structure of the four verses in the first strophe to that of random sequences of length 35 in Table 6 (when the encoding is with 3 letters *H*, *E* and *C*) and in Table 7 (when the encoding is with 4 letters *H*, *E*, *C* and *A*). Adverbs are too rare in verses of such a small length so that we did not considered the 5-letter case.

Let us first look at the 3-letter case in Table 6. Apart from the first verse in the strophe, the structure of the poem is very close to that of $F_2$, up to the index 6 (for the second verse) and up to the index 7 (for verses 3 and 4). Higher order indices could not be reached in our calculations. For the English translation, the closeness to $F_2$ holds as well but is not so perfect. It is not so surprising since the poem was originally composed in French. For a French translation of a poem in English one would have obtained a similar (small)

discrepancy to the group structure to $F_2$. We looked at the cardinality structure of the cc of subgroups by taking random sequences of length 35 in 10 runs and we observe that the closeness to $F_2$ is much less than in the case of the poem.

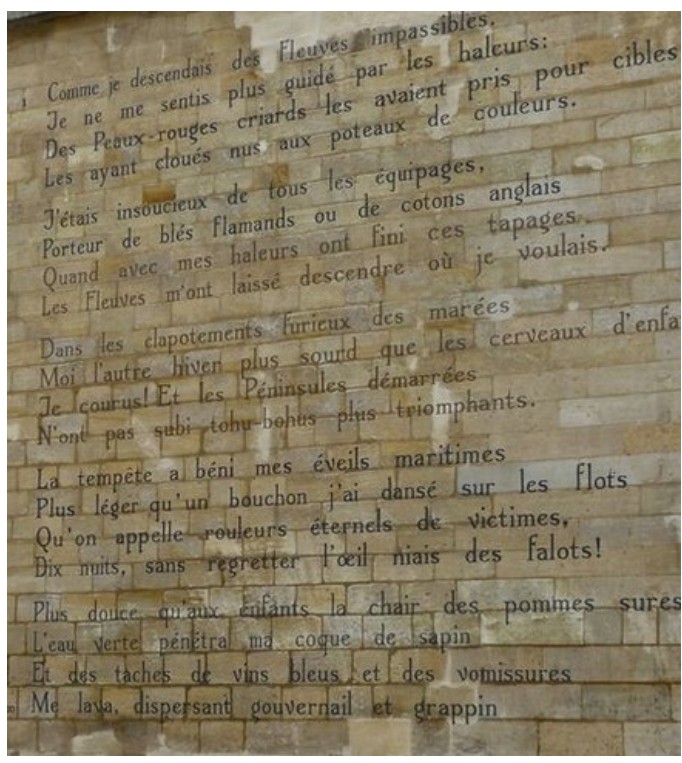

**Figure 4.** Part of the poem *Le Bateau Ivre* of Arthur Rimbaud on a wall (Rue Férou) in Paris.

**Table 6.** Group structure of the poem *Le Bateau Ivre'* (*The Drunken Boat*) by Arthur Rimbaud. Only the first strophe (that has four lines) is analyzed, firstly in its original form, then in an English translation. Each line is split into segments encoded by the symbol $H$ (for names and adjectives), $E$ (for verbs) or $C$ (for the other types: conjunctions, adverbs, prepositions, punctuation marks and so on). The group relation is displayed for the first line only.) The cardinality structure of cc of subgroups of a small index is compared to the one obtained with 10 runs of a sequence of random 3-letter words of similar length (i.e., the length 35).

| | | |
|---|---|---|
| Comme je descendais des fleuves impassibles,<br>rel=$C^4C^2E^{10}C^3H^7H^{11}C$ | [**1,1,7**,17,114, 1395,36973] | 1 |
| Je ne me sentis plus guidé par les haleurs: | [**1,3,7,26,97, 624**,4171] | 2 |
| Des Peaux-Rouges criards les avaient pris pour cibles | [**1,3,7,26,97, 624**,4163] | . |
| Les ayant cloués nus aux poteaux de couleurs. | [**1,3,7,26,97, 624**,4163] | . |
| As I was floating down unconcerned rivers<br>rel=$C^2 * C * E^3 * E^8 * C^4 * E^1 1 * H^6$ | [**1,3,7,26,97, 624,4163**,34470] | 2 |
| I now longer felt myself steered by the haulers: | [**1,3,7,26**,101, 656,4227] | 2 |
| Gaudy Redskins had taken them for targets | [**1,3,7,26,97, 624,4163**,324935] | . |
| Nailing them naked to coloured states. | [**1,3,7**,42,202, 1682,9204] | . |
| [Random[1,3]: i in [1..35]]<br>(10 runs) | [**1,3,7**,30, .] (×3)<br>[ **1,3,7,26**, . ]( ×3)<br>[**1,3,7**,.,.,]<br>[ **1,3,10**,.,. ](×2)<br>[ **1,3**,13,.,.] | 2<br>.<br>.<br>.<br>. |
| Isoc(X;2) | [**1,3,7,26,97, 624,4163**,34470] | 2 |

**Table 7.** The same as in Table 6, but each line is split into segments encoded by the symbol $H$ (for names and adjectives), $E$ (for verbs), $A$ for prepositions, or $C$ (for the other types: conjunctions, adverbs, punctuation marks and so on). The cardinality structure of cc of subgroups of a small index is compared to the one obtained with 10 runs of a sequence of random 4-letter words of similar length (i.e., the length 35).

| | | |
|---|---|---|
| Comme je descendais des fleuves impassibles, <br> rel=$C^4 C^2 E^{10} A^3 H^7 H^{11} C$ | **[1,7,41,604,13753]** | 3 |
| Je ne me sentis plus guidé par les haleurs: | **[1,7,41,604,13753]** | . |
| Des Peaux-Rouges criards les avaient pris pour cibles | **[1,7,41,604,13753]** | . |
| Les ayant cloués nus aux poteaux de couleurs. | **[1,7,41,604,13753]** | . |
| As I was floating down unconcerned rivers <br> rel=$C^2 C E^3 E^8 A^4 E^{11} H^6$ | **[1**,7,59,1386,27011] | 3 |
| I no longer felt myself steered by the haulers: | **[1,7,41,604,13753]** | . |
| Gaudy Redskins had taken them for targets | **[1**,7,50,1763,51582] | . |
| Nailing them naked to coloured states. | **[1**,7,59,1002,18671] | . |
| [Random[1,4]: i in [1..35]] <br> (10 runs) | **[1**,7,50,755,.] ($\times 2$) <br> **[1,7,41,604**,.] ($\times 3$) <br> [ **1,7,41**,.,.]($\times 2$) <br> [ **1,7**,50,739,.]($\times 2$) <br> [ **1,7**,59,.,.] | 3 <br> . <br> . <br> . <br> . |
| Isoc(X;3) | **[1,7,41,604,13753]** | 3 |

The group structure with 3 letters can also be obtained for the group structure with 4 letters in Table 7 but the closeness is to $F_3$ (not $F_2$), as expected.

## 6. Conclusions

The graph covering approach has been shown to be useful for understanding how complex structures are encoded in nature and in art. For proteins, there exists a primary encoding with 20 amino acids as letters and the secondary encoding determines the folding of proteins in the 3-dimensional space. This is useful for recognizing the relationship between the structure and function of the protein. We took examples based on a present hot topic: a variant of the SARS-Cov-2 spike protein and the alipoprotein-H. For music, the secondary structures are called musical forms and the choice of them determines the type of music. For poems, we took the French (or English) alphabet with 26 letters, but many other alphabets may be used for the application of our approach. The secondary structures are defined from the encoding of the words (names, verbs and so on).

It is also interesting to speculate about the possible existence of a primary code and a secondary code in other fields, for example, in physics at the elementary level like in particle physics and quantum gravity [35]. According to the experience of the authors of this paper, the structure has much to do with complete quantum information. The reader may consult paper [36] about particle mixings or [3,37] about the genetic code in which finite groups are the players. Here, we are dealing with infinite groups so that the representation theory of finite groups (with characters) has to be defined on finitely-presented groups (most of the time of infinite cardinality). This will be explored further in our next paper [38].

**Author Contributions:** Conceptualization, M.P., F.F. and K.I.; methodology, M.P. and R.A.; software, M.P.; validation, R.A., F.F. and M.M.A.; formal analysis, M.P. and M.M.A.; investigation, M.P., F.F. and M.M.A.; writing—original draft preparation, M.P.; writing—review and editing, M.P.; visualization, F.F. and R.A.; supervision, M.P. and K.I.; project administration, K.I.; funding acquisition, K.I. All authors have read and agreed to the published version of the manuscript.

**Funding:** Funding was obtained from Quantum Gravity Research in Los Angeles, CA.

**Institutional Review Board Statement:** Not applicable.

**Informed Consent Statement:** Not applicable.

**Data Availability Statement:** Not applicable.

**Conflicts of Interest:** The authors declare no conflict of interest.

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
