# Peer review of "Graph Coverings for Investigating Non Local Structures in Proteins, Music and Poems"

_sci, doi:10.3390/sci3040039_

Round 1
Reviewer 1 Report
The aim of this paper is to analyze a “secondary structure of proteins, musical forms and verses of poems [which] are approximately ruled by universal laws relying on graph coverings.”.
Relevance: very relevant
The paper is “very relevant” due to the following elements:
- the subject is very well defined and tangible
- the technical aspects are well justified and detailed
The paper is significant due to the following arguments:
- the authors' original contribution is very clearly explained
- the used methods are well described
From a quality point of view, the manuscript is technically correct. The entire manuscript is well written, but the template of the journal paper must be used. Meantime, new references can be added and a comparison with the obtained results vs data literature would be welcome.
Anyway, the paper can be used by any person desiring to become familiar with the field and to develop an interest in the area of graph application.
Author Response
Thank you for reading our work. We are glad that you found the paper could be published as is.
Following your advice we added a brief review of the literature at the end of the (new) introduction.
Some other changes were performed to account for the recommendation of the other referees.
The MDPI template was now used.
Reviewer 2 Report
This is a well written and engaging piece of research.
The subject of how complex structures are encoded in nature and in art is interesting.
In fact, I thank you for the work you have done. But perhaps, it would be convenient to see the possibility of looking for another journal to index this manuscript (editorial / subject line).
Nevertheless, I recommend this paper acceptance.
The conclusion is thorough.
The references, although varied, are up to date. Which says a lot.
Thank you for the opportunity to review this engaging and valuable paper. I wish you every success in your joint and individual future research projects.
Author Response
Thank you for reading our work. We are glad that you found the paper could be published as is.
Following the advice of the other referees, we added an introduction, a brief review of the literature and a new abstract.
Reviewer 3 Report
This paper deals with the secondary structure of proteins, musical forms, and verses of poems that are approximately ruled by universal laws relying on graph coverings.
The abstract should state briefly the purpose of the research, the principal results, and major conclusions. An abstract is often presented separately from the article, so it must be able to stand alone.
The author must use the template for the journal paper.
The authors can improve the literature review with news references: N.Shakhovska, S. Fedushko. Data Analysis of Music Preferences of Web Users based on Social and Demographic Factors. Procedia Computer Science (in press).
Consider the length of the conclusions and the authors should check the results against similar works in the discussion section. It is suggested to compare the results of the present research with some similar studies which is done before.
Add DOI for all references.
The paper should be divided into the following sections: 1. Introduction, 2. Materials and Methods, 3. Results, 4. Discussion, 5. Conclusions
The paper needs to be better reorganized also in subsections, with better and easier differentiation.
The way the data is collected lacks validity and reliability. The authors explicitly expressed this problem in the "Limitations" of the study, but I still think it is a very problematic issue.
It is suggested to organize the Conclusion section much better.
Thank you for a good job.
Author Response
Thank you for refering about our work.
We took account for most of your recommendations; They were very useful for improving the quality of the paper.
* a new abstract with the purpose of the research, the results and conclusions,
* we used the MDPI template,
* an introduction with a brief review of the literature (including the suggested reference): this a way to reorganize the paper since now the
mathematical part is Section 2. We found necesary to keep the remaining three topics (proteins, musical forms and poetry) in three separate sections,
* but the conclusion was left almost unchanged. Its role is a brief overview of what are the perspectives for future research.
Reviewer 4 Report
- Abstract is very confusing and needs major revision. The abstract should address the problem (what), significance (why), approach (how), and overall findings. I do not see any of that.
- Also, you need a reference for the first sentence.
- What is graph covering?
- The introduction section is missing the actual “introduction” of the research. You would want to start with a short background of the problem, its significance, why you are doing this, and then the research hypothesis/objectives/contributions.
- You need to define some of the terms up front before using them. For example: graph covering, secondary structure, etc.
- The literature review is missing. What has been done in the past, what is the gap you are trying to cover? Why is your approach better than the current ones in literature? How do you quantify the improvements in your approach compared to the previous ones?
- This paper is missing a formal structure and it should include the following at the minimum: Introduction, Literature Review, Methodology, Analysis/Results, Discussion/Conclusion
- How do you quantify your contributions? How is it better from the previous work?
Author Response
Thank you for refering about our work.
We took account for most of your recommendations. They were very useful for improving the quality of the paper.
* There is a new informative abstract,
* the meaning of graph covering is explained in Section 2 (the mathematical section),
* there are new references at the appropriate places
* there is an introduction (before the mathematical section) with a short review of the literature. It is explained why our approach encompasses
standard work.
* We think that the reorganization: Conclusions, Mathematics and three extra sections for the three topics (proteins, musical forms and poetry), Conclusion
is the best way to organize our findings.
Round 2
Reviewer 3 Report
Accept in present form
Author Response
Thank you again for your reading and recommendations.
Reviewer 4 Report
- Abstract is still not clear. It starts with “We explore the structure similarities in…,” but what is the problem? and why is so important? And what does that achieve? What is the impact? Then you can explain what you are doing to address this problem. Also, you need to talk about your approach (how you are doing this).
- What do you mean by the referring to protein as language and its letters as amino acid? You need to clarify why/how you came up with this analogy.
- Define local vs non-local letters.
- Page 2, Lines 56-60 seem to be unnecessary information. Either remove or revise and make it shorter and to the point.
- You keep using the clause “our conclusion is that these structures are similar” but you did not explain how you define and measure the similarity.
- There should be a section after the Methodology to summarize the findings and how this finding can be applied and contribute to science.
- How do you quantify your contributions? How is it better from the previous work?
Author Response
Thanks again for looking at the paper. Your first report was very useful for us in preparing this second version. We did a strong effort
in accounting for your recommendations.We cannot do more for the following reasons that we explain point by point.
“We explore the structure similarities in…,”: the second sentence states that the similarities are in the group structure, more precisely in the closeness to the free group $F_{r-1}$.
"Also, you need to talk about your approach (how you are doing this)", in our view the abstract is intended to summarize not to give all details. The (mathematical) section 2 deals about the how.
"What do you mean by the referring to protein as language and its letters as amino acid? You need to clarify why/how you came up with this analogy.":
this is clearly explained in the introduction, lines 20-36.
"Define local vs non-local letters." Again, it is explained in the introduction that we pass from 20 letters/amino acids to 3 letters (encoding coils, $\alpha$ helices
and $\beta$- sheets) or more letters in Section 3.
"Page 2, Lines 56-60 seem to be unnecessary information. Either remove or revise and make it shorter and to the point."
It is strange that the referee rejects this sentence because it replies to his own comment: why we did the comparison between the 'language' of proteins and poetry language.
We revise the sentence by removing "with F.F. Nichita as a Guest Editor" and "immediately".
"You keep using the clause “our conclusion is that these structures are similar” but you did not explain how you define and measure the similarity."
Again we refer to the beginning of this response.: the second sentence states that the similarities are in the group structure, more precisely in the closeness to the free group $F_{r-1}$.
The whole paper is intendes to proove this mathematical claim. Please read the paper again. We are sad if you are not convinced!
"There should be a section after the Methodology to summarize the findings and how this finding can be applied and contribute to science.":
this is already summarized in the abstract, in the introduction and the conclusion.
" How do you quantify your contributions? How is it better from the previous work?", this paper is available as a Preprint and on HAL. It already has a good impact.
Look at our review of the literature to see why it is different and (possibly) better than previous work.
Round 3
Reviewer 4 Report
None of my previous comments have been addressed in this version. So, I am providing them here again:
- Abstract is still not clear. It starts with “We explore the structure similarities in…,” but what is the problem? and why is so important? And what does that achieve? What is the impact? Then you can explain what you are doing to address this problem. Also, you need to talk about your approach (how you are doing this).
- What do you mean by the referring to protein as language and its letters as amino acid? You need to clarify why/how you came up with this analogy.
- Define local vs non-local letters.
- Page 2, Lines 56-60 seem to be unnecessary information. Either remove or revise and make it shorter and to the point.
- You keep using the clause “our conclusion is that these structures are similar” but you did not explain how you define and measure the similarity.
- There should be a section after the Methodology to summarize the findings and how this finding can be applied and contribute to science.
- How do you quantify your contributions? How is it better from the previous work?